# Clinical Evaluation and Comparison of Two Microfluidic Antigenic Assays for Detection of SARS-CoV-2 Virus

**DOI:** 10.3390/microorganisms11112709

**Published:** 2023-11-05

**Authors:** Paolo Bottino, Valentina Pizzo, Salvatore Castaldo, Elisabetta Scomparin, Cristina Bara, Marcella Cerrato, Sabrina Sisinni, Serena Penpa, Annalisa Roveta, Maria Gerbino, Antonio Maconi, Andrea Rocchetti

**Affiliations:** 1Microbiology and Virology Laboratory, A.O. “SS Antonio e Biagio e C. Arrigo”, Via Venezia 16, 15121 Alessandria, Italy; valentina.pizzo@ospedale.al.it (V.P.); castaldosal90@gmail.com (S.C.); escomparin@ospedale.al.it (E.S.); cbara@ospedale.al.it (C.B.); marcella.cerrato@ospedale.al.it (M.C.); arocchetti@ospedale.al.it (A.R.); 2Research and Innovation Department (DAIRI), A.O. “SS. Antonio e Biagio e C. Arrigo”, Via Venezia 16, 15121 Alessandria, Italy; sabrina.sisinni@ospedale.al.it (S.S.); serena.penpa@ospedale.al.it (S.P.); aroveta@ospedale.al.it (A.R.); amaconi@ospedale.al.it (A.M.); 3Department of Science and Technological Innovation (DISIT), University of Eastern Piedmont, Viale Teresa Michel 11, 15121 Alessandria, Italy; mary.gerbino99@gmail.com

**Keywords:** SARS-CoV-2, COVID-19, third-generation assay, microfluidic

## Abstract

Given the ongoing pandemic, there is a need to identify SARS-CoV-2 and differentiate it from other respiratory viral infections in various critical settings. Since its introduction, rapid antigen testing is spreading worldwide, but diagnostic accuracy is extremely variable and often in disagreement with the manufacturer’s specifications. Our study compared the clinical performances of two microfluidic rapid antigen tests towards a molecular assay, starting from positive samples. A total of 151 swabs collected at the Microbiology and Virology Laboratory of A.O. “SS Antonio e Biagio e C. Arrigo” (Alessandria, Italy) for the diagnosis of SARS-CoV-2 were simultaneously tested to evaluate accuracy, specificity, and agreement with the RT-qPCR results. Both assays showed an overall agreement of 100% for negative specimens, while positive accuracy comprised between 45.10% and 54.90%. According to the manufacturer’s instructions, the greatest correlation between the antigenic and molecular assays was observed for the subset with high viral load (18/19, 94.74%), while it dramatically decreased for other subsets. Moreover, the ability to differentiate between SARS-CoV-2 and Flu provides an added value and could be addressed in an epidemic context. However, an in-house validation should be performed due to differences observed in performance declared by manufacturers and those actually obtained.

## 1. Introduction

Coronavirus disease (COVID-19) is an infectious disease caused by severe acute respiratory syndrome coronavirus 2 (SARS-CoV-2), a single-stranded positive-sense RNA virus with four main structural proteins: spike, envelope, membrane, and nucleocapsid [1]. It belongs to the human coronavirus family (HCoV), of which six were already known to cause disease. Among these, four are known as human endemic coronaviruses (HCoV-229E, HCoV-NL63, HCoV-OC43, HCoV-HKU1) responsible for acute self-limiting common-cold symptoms, while the other two (SARS-CoV and MERS-CoV) cause outbreaks of severe lower respiratory tract infection [2]. A quick and accurate detection of SARS-CoV-2 infected individuals is important to minimize the spread of COVID-19. However, it immediately represented a substantial challenge for healthcare systems [3]. Qualitative reverse transcription qPCR (RT-qPCR) testing of specimens collected at a level of the upper respiratory tract (nasopharyngeal and oropharyngeal secretion) was the first diagnostic approach developed for reliable detection of viral RNA, and it is still considered the gold standard for COVID-19 diagnosis [4]. Nevertheless, molecular testing is expensive, time consuming and requires adequately skilled staff; moreover, it can take several hours (average 3–4 h) to obtain a result. On the other hand, rapid antigen testing for the detection of SARS-CoV-2 is spreading worldwide, allowing a quick diagnosis in many different community settings; however, diagnostic accuracy is extremely variable, with sensitivity ranging from 0% to 94% [5]. Rapid tests are based on lateral flow immunochromatography using antibodies against SARS-CoV-2 proteins (antigens), present in respiratory tract specimens, and detected mainly the viral nucleoprotein, less frequently the spike proteins [6]. Nonetheless, several immunoassays revealed an overall diagnostic sensitivity that settled at around 70%, lower than the minimum diagnostic sensitivity (≥80%) required by the Task Force on COVID-19 (International Federation of Clinical Chemistry and Laboratory Medicine, IFCC) and World Health Organization (WHO) [7,8]. In addition to manual lateral flow immunoassays (first and second generation) and laboratory-based chemiluminescent tests (fourth generation), the microfluidic assays (third generation), of which the LumiraDx SARS-CoV-2 Ag test represents the prototype for rapidity, handiness, and versatility as a decentralized testing device, could emerge as a reliable diagnostic alternative [9]. Since not only SARS-CoV-2 but also influenza viruses A and B are highly transmissible and share some overlapping signs and symptoms (cough, sore throat, fever, headache, respiratory distress), a quick and accurate differential diagnosis represents a pivotal step in order to prevent further spread of the viral disease and providing a suitable treatment [10]. For these reasons, combo antigen tests have been developed for accurate diagnosis of SARS-CoV-2 and differentiation from other respiratory infections. Although the analytical performance (i.e., accuracy, sensitivity, and specificity) of these newly introduced tests have been determined in manufacturers’ studies, they may diverge in routine clinical practice and in their use among the general population [11]. Hence, considering the biological and technical features, an in situ evaluation should be performed at different hospital settings before introducing a new antigenic assay for SARS-CoV-2 detection.

Thus, due to the abovementioned aspects, the aim of our study was a retrospective analysis focused on clinical performances of the antigenic tests LumiraDx SARS-CoV-2 & Flu A/B (LumiraDx Limited, UK) and LumiraDx SARS-CoV-2 Ag Ultra (LumiraDx Limited, UK) for detection of SARS-CoV-2, starting from clinical samples collected in viral transport medium (VTM) with different viral loads.

## 2. Materials and Methods

A total of 151 nasopharyngeal swabs were collected and stored at Microbiology and Virology Laboratory of A.O. “SS Antonio e Biagio e C. Arrigo” (Alessandria, Italy) for diagnosis of SARS-CoV-2 infection. All samples were tested according to WHO guidelines [12] and collected in 3 mL of liquid universal transport medium (UTM) (Copan, Italy). Each specimen was assessed for detection of SARS-CoV-2 via RT-qPCR (Xpert^®^ Xpress SARS-CoV-2, Cepheid, Sunnyvale, CA, USA) used as standard of care (SoC).

Furthermore, clinical specimens were assigned to four groups: negative SARS-CoV-2 (100 samples), very high viral load (Ct value 17–25; 19 samples), high viral load (Ct value 25–30; 23 samples), and moderate viral load (Ct value 30–36; 9 samples).

All samples were simultaneously tested with antigenic tests LumiraDx SARS-CoV-2 & Flu A/B and LumiraDx SARS-CoV-2 Ag Ultra according to manufacturer instructions.

The LumiraDx immunoassays herein reported were based on rapid microfluidic immunofluorescence for qualitative detection of SARS-CoV-2 Nucleocapsid proteins (LumiraDx SARS-CoV-2 Ag Ultra) and Influenza Type A/B Nucleocapsid proteins (tests LumiraDx SARS-CoV-2 & Flu A/B) in order to differentiate the etiological agents of viral disease. The assay principle relied on fluorescent latex nanoparticles coated with monoclonal antibodies and magnetic beads targeted to the abovementioned viral antigenic structures.

Briefly, 100µL of UTM was transferred in a vial and collected with a swab until complete retention. Subsequently, the soaked swab was transferred into extraction buffer, loaded onto test strip for both tests, and analysed with LumiraDx instrument according to the manufacturer’s instructions. Within 12 min, the test result (positive or negative) was reported.

For each specimen group, both antigenic methods were compared in terms of accuracy, specificity, and agreement towards the molecular SoC and between them.

## 3. Results

According to RT-qPCR, on a total of 151 samples included in our analysis, 51 samples (33.8%) tested positive, while 100 (66.2%) were negative. On positive samples, the overall median Ct was 27.5 ± 5.99 (IQR: 19.8–30.0). According to the abovementioned criteria, specimens with detected viral load were divided into three groups as follows: 19 samples at very high viral load (Median Ct: 18.5 ± 2.69, IQR: 17.0–20.5), 23 at high viral load (Median Ct: 28.5 ± 1.69, IQR: 27.3–30.0), and 9 with moderate viral load (Median Ct: 34.5 ± 0.89, IQR: 34.0–34.5) (Table 1).

Looking at the diagnostic accuracy of LumiraDx SARS-CoV-2 Ag Ultra towards the SoC, the specificity for negative samples was 100%, while the accuracy was 54.90 (CI: 40.34% to 68.87%). The overall agreement was 0.62 (CI: 0.48 to 0.75). Focusing on samples with very high viral load accuracy and inter-rater agreement were, respectively, 94.74% (CI: 73.97% to 99.87%) and 0.97 (CI: 0.90 to 1), while those with high viral load decreased (accuracy: 43.48, CI: 23.19% to 65.50%; k-value: 0.56, CI: 0.35 to 0.76). No samples with moderate viral load yielded a positive result for the antigenic test (Table 2 and Figure 1).

Instead, focusing on the diagnostic accuracy of LumiraDx SARS-CoV-2 & Flu A/B towards the SoC, the specificity for negative samples was 100%, while the accuracy was 45.10 (CI: 31.13% to 59.66%). The overall inter-rater agreement was 0.52 (CI: 0.38 to 0.66). For samples with very high viral load accuracy and agreement were, respectively, 94.74% (CI: 73.97% to 99.87%) and 0.97 (CI: 0.90 to 1) and more reduced for those with high viral load (accuracy: 21.74, CI: 7.46% to 43.70%; k-value: 0.31, CI: 0.09 to 0.52). Again, no samples with moderate viral load yielded a positive result for the antigenic test compared with RT-qPCR (Table 2 and Figure 1).

The inter-rater agreement of LumiraDx SARS-CoV-2 Ag Ultra and SARS-CoV-2 & Flu A/B resulted in 0.88 (CI: 0.77 to 0.98). A complete agreement was obtained for samples with high viral load (k-value: 1), while for those with high viral load, the inter-rater agreement was 0.64 (CI: 0.36 to 0.93) (Table 3).

## 4. Discussion

In this study, we determined the performance characteristics of the two assays, the LumiraDx SARS-CoV-2 Ag Ultra and LumiraDx SARS-CoV-2 & Flu A/B, for the detection of SARS-CoV-2 virus in respiratory samples and compared the results with RT-qPCR, used as reference.

Since their introduction in autumn 2020, antigenic tests have become a key part of testing strategies, thus suggesting their deployment in order to detect potential infectivity and help control the spread of infection rather than for the purpose of clinical diagnosis [6]. Despite rapid antigen tests reported to have good performances (agreement >90% for positive RT-qPCR results), a wide range of clinical accuracy, often lower than declared by the manufacturers, was observed in clinical samples [13]. A comparison of several commercial lateral flow/chemiluminescent assays showed values ranging from 20.0% to 90.0% [14,15,16,17], nearly always lower than the value reported in the kits IFU. Herein, we observed overall accuracies of 54.90% and 45,10% for LumiraDx SARS-CoV-2 Ag Ultra and LumiraDx SARS-CoV-2 & Flu A/B, respectively. However, focusing on a subset of samples with higher viral load (cycle threshold range: 17–25), this value amounted to 94.74% for both methods, more than what was declared by the manufacturer (LumiraDx SARS-CoV-2 Ag Ultra: 92.7%; LumiraDx SARS-CoV-2 & Flu A/B: 95.5%). In the range of 25–30 cycle threshold, the diagnostic accuracy dramatically decreased, especially for the SARS-CoV-2/Flu A/Flu B assay (21.74%). Both kits appeared not able to detect positive samples with lower viral loads (Cycle threshold > 30). Considering that individuals with fever and respiratory symptoms showed higher nasopharyngeal SARS-CoV-2 viral loads in terms both of digital PCR (Viral load quantification) and qPCR (Ct values) than those without those symptoms [18,19], the antigenic tests abovementioned might provide a useful assay to screen symptomatic patients in critical settings. Moreover, the possibility to use the same instrument to read results obtained by both kits may provide the ability to rapidly switch between them according to the pandemic context (i.e., presence or absence of contemporary Flu peak and increase in SARS-CoV-2 infection) [20].

Interestingly, we observed a higher accuracy for LumiraDx SARS-CoV-2 Ag Ultra towards LumiraDx SARS-CoV-2 & Flu A/B, despite the declared limit of detection of this latter was lower (80 TCID50/mL vs. 800 TCID50/mL for LumiraDx SARS-CoV-2 Ag Ultra). This could probably be due to the different SARS-CoV-2 strains used for the evaluation of the test in terms of isolate and supplier. In our opinion, according to the real performance observed in clinical settings of those declared [6], this stressed the need for in-house validations of antigenic assays in order to provide the most suitable testing strategy. Thus, we should consider the added value of these assays in terms of cost, rapidity, and impact on the clinical management of symptomatic patients.

In this study, we focused on the antigenic detection of the SARS-CoV-2 virus, not evaluating the interaction with Flu A or B or other respiratory viruses. Moreover, due to the small sample size, no impact of variants on clinical performances was assessed. Finally, it is noteworthy that our results were obtained via UTM samples tested according to the manufacturer’s instructions, not directly from collected swabs. Thus, there is a dilution factor to be considered. Future analysis will be addressed to evaluate the relationship between the two assays in clinical settings, patients’ symptomatology, and epidemic context.

In summary, rapid antigenic tests could provide useful support to molecular testing, especially in critical settings, but the advantages and limitations should be evaluated via an in-house analysis.

## Figures and Tables

**Figure 1 microorganisms-11-02709-f001:**
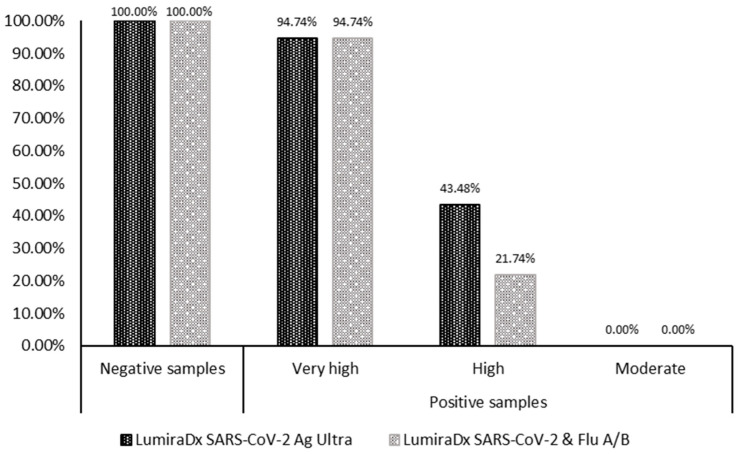
Specificity and accuracy of the two microfluidic antigenic tests according to subsets samples.

**Table 1 microorganisms-11-02709-t001:** Descriptive analysis of tested samples.

	Positive Samples	Negative Samples	Total			
N. (%)	51 (33.8%)	100 (66.2%)	151 (100%)			
Ct Values of positive samples						
	Mean	Median	Standard Deviation (SD)	Interquartile Range (IQR)	Min	Max
Overall	26.0	27.0	5.9	10.3	17.0	35.0
Very high	19.2	18.5	2.7	3.5	17.0	25.0
High	28.4	28.5	1.7	2.7	25.5	30.0
Moderate	34.1	34.5	0.9	0.5	32.0	35.0

**Table 2 microorganisms-11-02709-t002:** Diagnostic performances of the two microfluidic antigenic tests.

	Accuracy	Specificity	Cohen’s Kappa Coefficient
LumiraDx SARS-CoV-2 Ag Ultra vs. RT-qPCR			
Overall	54.90%	100.00%	0.62
Very high	94.74%		0.97
High	43.48%		0.56
Moderate	0.00%		0.00
LumiraDx SARS-CoV-2 & Flu A/B vs. RT-qPCR			
Overall	45.10%	100.00%	0.52
Very high	94.74%		0.97
High	21.74%		0.31
Moderate	0.00%		0.00

**Table 3 microorganisms-11-02709-t003:** Comparison of diagnostic performance between the two antigenic tests.

	SARS-CoV-2 Ultra (N. Pos)	SARS-CoV-2 & Flu A/B (N. Pos)	Cohen’s Kappa Coefficient
Overall	28	23	0.88
Very high	18	18	1.00
High	10	5	0.64
Moderate	0	0	0.00

## Data Availability

The data contained in this manuscript are available upon request.

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
