# Peer review of "Clinical Evaluation and Comparison of Two Microfluidic Antigenic Assays for Detection of SARS-CoV-2 Virus"

_microorganisms, 2023, doi:10.3390/microorganisms11112709_

Round 1
Reviewer 1 Report
Comments and Suggestions for Authors
This paper evaluates the clinical performance of two microfluidic rapid antigen tests against a molecular assay using positive samples. 151 swabs collected in Alessandria, Italy, were tested to determine their sensitivity, specificity, and agreement with RT-qPCR results for diagnosing SARS-CoV-2. Overall, the paper is well-written and organized. However, there are a few suggestions for the authors:
1. The introduction section does not highlight the novelty of this research.
2. The paper lacks insights from the device perspective. For instance, the authors should explain the detection principle of the two microfluidic antigen assays.
Author Response
We are very grateful for the advices and the comments provided by the reviewers in order to improve our manuscript. We have taken in account all these corrections in the revised version of the manuscript. Below I have stated our response to each of the suggestions made by the reviewers. We hope that the modifications made in the revised manuscript and that the way we addressed the comments made by you and the reviewers meet your expectations.
Sincerely
- The introduction section does not highlight the novelty of this research.
We agree with the reviewer for the suggestion proposed. We reformulated the final part of introduction (Lines 70-77) in order to link the reason for which should be performed an in-situ evaluation and our goal. According to the literature reported and also our experience, it is important to define the real clinical performance of each antigenic assay before introducing it in clinical routine. Despite it wasn’t a novelty, the take-home message of our manuscript would like to highlight this need.
- The paper lacks insights from the device perspective. For instance, the authors should explain the detection principle of the two microfluidic antigen assays.
According to reviewer suggestion, we added a short paragraph with some details about microfluidic principle of tested assays (Line 90-96).
All the changes mentioned and reported in the text have been highlighted alongside the manuscript.
Reviewer 2 Report
Comments and Suggestions for Authors
Dear Editor and Authors,
I have read the paper entitled "Clinical evaluation and comparison of two microfluidic antigenic assays for detection of SARS-CoV-2 virus", which deals with an interesting and relevant topic that could be attractive for Readers interested in both the manufacturing of SARS-CoV-2 detection kits, as well as those using them in clinical practice.
Content-wise, the paper is generally excellently written (it was a pleasure to read it!), relatively well structured, and with a fluent reasoning for explaining the performed analysis and the drawn conclusions. However, as I slowly advanced thru the paper, I realized that the paper still has a few important issues that need to be addressed, hence my recommendation for MAJOR corrections. All the required clarifications/corrections are detailed below.
MAJOR
- In the Abstract, it is indicated first that the detection sensitivity for positive patients was between 45.1% and 54.9% for both tested assays. I believe there are 2 critical problems raised by this statement.
First, in my opinion the next statement (p.1, at lines 25-26) "The microfluidic antigenic tests showed an overall excellent clinical performance" is in total contradiction with the previous statement. The previously indicated numbers basically show that the assays have an average probability of about 50% false negative diagnosis, which I believe it is very bad, particularly for a highly infectious and potentially dangerous disease as COVID (see also the end of the next paragraph).
Second, the first statement also points out that the parameter whose value was indicated in it (and also repeatedly throughout the paper, see for instance p.2 lines 49 and 53-55 and in many other places) is actually NOT the sensitivity. For the correct general definitions of the typical performance characteristics (sensitivity, resolution, (in)accuracy, limit of detection, linearity, errors -which can be systematic or random, etc.) of a sensor of ANY type, I strongly suggest the Authors to consult the "Handbook of Modern Sensors" by Jacob Fraden (2016), Chapter 2 "Transfer Functions" p.13-34, and especially Chapter 3 "Sensor Characteristics" p.35-68. They will see there that the correct general definition of sensitivity S is that S is the slope in a certain point of the transfer characteristic which is the plot of sensor output parameter (let's denote it here by "POut") versus the sensor's input parameter (let's denote it here by "Pin"), i.e. S = dPOut/dPin i.e. S is the slope in one single point of the graph showing the variation of POut vs. Pin. In the specific case where this dependency POut = f(Pin) is linear, then S is indeed the slope of the entire line of the plot. Hence, in our case, the sensor/assay sensitivity output parameter should be -in my opinion- given by the colour and/or (colour) intensity change of the assay (which should be accurately measured quantitatively using a highly sensitive and accurate calibrated device such as a colorimeter or optical detector) while the input parameter is clearly the chemical concentration of the SARS-CoV-2 antigen(s) in the blood. Once the plot showing the dependence of the former on the latter is charted, then the actual sensitivity can be easily deduced if the dependency is indeed linear (which frankly I doubt very much, given the results shown by the Authors after their analysis). What the Authors indicated in their statement(s) is actually (what is defined in Fraden's book as) the accuracy, expressed in terms of percentage of the input span (full scale). Moreover, as I already mentioned previously, even in this case the quantitative values given there (in the Abstract) are not good enough to justify widespread medical usage of those assays for diagnosis, as is already acknowledged by the Authors themselves in the paper, at p.2 lines 54-55. Therefore, the statement about the "excellent clinical performance" in the Abstract should be deleted, or -at best- drastically rephrased. This may also require some rephrasing of some of the conclusions as well. Thus, the statement "individuals with fever and respiratory symptoms showed higher viral loads than those without those symptoms" should be explained and justified in more detail, so as to demonstrate a clear dependence, or at least some correlation, which can clearly justify the rapid decrease of performance with the viral load as practically acceptable.
Likewise, the term stated as sensitivity in Table 2 is again actually accuracy, and the reference in the text of "diagnostic accuracy" (p.3 lines 103 and 110) should be deleted and the respective sentences rephrased, e.g. "The performance of the LumiraDX .... kit shows an accuracy of ... and a specificity of ...", and similarly the usage of "sensitivity" in other places (e.g. p.3 lines 113 and 115 but also later in other places in the paper which -for brevity- I will not indicate them here one by one) should be sanctioned and correspondingly modified as well.
- The expression of the high viral load which (at p.1 line 24) is first given in percentages (percentages of what?? What is the reference?) should rather be made in terms of the input parameter, i.e. the concentration of SARS-CoV-2 antigens in the blood. This percentage expression is also different from the "viral load" introduced at p.2 lines 83-84, but even here the Authors should briefly define/explain how this viral load is defined and especially measured, in what units?? Also, I personally find it a bit puzzling why the viral load is denoted by "Ct" as none of its letters seem to have any correspondence with the expression "viral load", and by the way shouldn't "t" be a subscript?? Or, in fact, like for many (bio)chemical sensors, "Ct" stands for total concentration (of the measured biocomponent), expressed in, e.g., [milimol/militer]?? Again, the Authors should clarify this aspect as well. Personally, I believe that the Authors should provide an Additional Material (in a separate file, to be added in the Journal together with the main paper) in which they can either briefly describe the definition of all other parameters (e.g. specificity, CI, Cohen's kappa coefficient, k-value, etc.) and/or provide some references and links where (more) such info is provided. This would be especially helpful for readers who are not yet fully familiarized to the domain, with the best example being Ph.D. students who are just starting their work in this field.
- Finally, allow me to make an observation which (at least in my personal opinion, of course) could potentially strengthen significantly the paper. I couldn't help noticing that the paper was very short, and this led me to a more detailed pondering of a few issues which I detail in the next paragraphs.
It is unclear for the Reader WHY the Authors chose those specific LumiraDx products. Is it based on cost? or on what? Definitely they cannot be the only products available for the same purpose. I believe that there must be (many) more such products, and then it could be highly relevant for potentially a larger range of Readers, for the Authors to make a similar comparison of a few other representative devices as well. Hence, my suggestion would be for the Authors to add another section, in which they should:
- First, just summarize the (majority of the) most significant products present on the market and available for the same purpose. Of course, some more literature review may also be required to check and present if/what other similar reviews/product comparisons may have been done in literature until now.
- Second/Next, from all those products the Authors should then make a selection of only a few (1, 2, 3? or maybe max.4) of them (i.e. besides those already analyzed in the paper, but whose choice should also be justified, as I mentioned previously). This selection/choice MUST be clearly justified and explained and it must be based -again- on a very clear judgment/selection criterion, e.g.: most popular? most performant? cheapest? Or one (the best) of each category??. Hence a "metric", or a quantitative parameter that describes/characterizes each product of ALL those previously summarized should preferably be defined (or selected from the product's specs) and used to rank it in a list or in more lists (one for each parameter or producer), and then the Authors could presumably select the leader product at the top of each such list. Once these a few other products/devices are thus selected, then the Authors can further carry out the same type of analysis which they did for the LumiraDx kits. Yes, this would mean MUCH more work and more time to spend with experiments and analyses of their results for the benefit of the Readers (eh, noblesse oblige...), but I hope the Authors would agree that such an addition will not only strengthen considerably both the originality and the quality of their paper, but -due to its generality (no longer dealing only with just 2 products from the same manufacturer but with the key products from the major players on the market)- would drastically boost its practical relevance and also its attractiveness for a (much?) larger audience (possibly additionally including more important decisional factors in medical care institutions!).
MINOR
- ANY Figure or Table should not be located in the middle of a page and interrupt the fluency of the text, but all Figures/Tables on a page should be placed either at its top and/or its bottom or occupy a whole page.
- A minor error that may need to be corrected is at p.1 lines 17-18, in the Abstract: replace "non according to manufacturer declarations" with "in disagreement with the manufacturer's specifications".
- A personal suggestion is that some graphics should be added; for many people a drawn graph is more evocative than the numbers in the given Tables. Hence, I suggest the Authors to consider plotting in, e.g., bar charts the data shown in the Tables given in the paper, at least for Tables 1 & 2.
Once all these corrections/additions are done, I believe that the Authors can forward the revised paper directly to the Editor who can decide its publication.
With best wishes,
The Referee
Comments on the Quality of English LanguageI already indicated that English language and style are fine/minor spell check required. Also see the other comments/observations in the texts in the other sections for Authors & Editors.
Author Response
Dear Editor,
We are very grateful for the advices and the comments provided by the reviewers in order to improve our manuscript. We have taken in account all these corrections in the revised version of the manuscript. Below I have stated our response to each of the suggestions made by the reviewers. We hope that the modifications made in the revised manuscript and that the way we addressed the comments made by you and the reviewers meet your expectations.
Sincerely,
MAJOR
- In the Abstract, it is indicated first that the detection sensitivity for positive patients was between 45.1% and 54.9% for both tested assays. I believe there are 2 critical problems raised by this statement.
- First, in my opinion the next statement (p.1, at lines 25-26) "The microfluidic antigenic tests showed an overall excellent clinical performance" is in total contradiction with the previous statement. The previously indicated numbers basically show that the assays have an average probability of about 50% false negative diagnosis, which I believe it is very bad, particularly for a highly infectious and potentially dangerous disease as COVID (see also the end of the next paragraph).
The abstract was reformulated deleting the abovementioned statement and focusing on excellent performances for subset with very high viral load, the one according with manufacturer declarations (Line 24-25).
- Second, the first statement also points out that the parameter whose value was indicated in it (and also repeatedly throughout the paper, see for instance p.2 lines 49 and 53-55 and in many other places) is actually NOT the sensitivity. For the correct general definitions of the typical performance characteristics (sensitivity, resolution, (in)accuracy, limit of detection, linearity, errors -which can be systematic or random, etc.) of a sensor of ANY type, I strongly suggest the Authors to consult the "Handbook of Modern Sensors" by Jacob Fraden (2016), Chapter 2 "Transfer Functions" p.13-34, and especially Chapter 3 "Sensor Characteristics" p.35-68. They will see there that the correct general definition of sensitivity S is that S is the slope in a certain point of the transfer characteristic which is the plot of sensor output parameter (let's denote it here by "POut") versus the sensor's input parameter (let's denote it here by "Pin"), i.e. S = dPOut/dPin i.e. S is the slope in one single point of the graph showing the variation of POut vs. Pin. In the specific case where this dependency POut = f(Pin) is linear, then S is indeed the slope of the entire line of the plot. Hence, in our case, the sensor/assay sensitivity output parameter should be -in my opinion- given by the colour and/or (colour) intensity change of the assay (which should be accurately measured quantitatively using a highly sensitive and accurate calibrated device such as a colorimeter or optical detector) while the input parameter is clearly the chemical concentration of the SARS-CoV-2 antigen(s) in the blood. Once the plot showing the dependence of the former on the latter is charted, then the actual sensitivity can be easily deduced if the dependency is indeed linear (which frankly I doubt very much, given the results shown by the Authors after their analysis). What the Authors indicated in their statement(s) is actually (what is defined in Fraden's book as) the accuracy, expressed in terms of percentage of the input span (full scale). Moreover, as I already mentioned previously, even in this case the quantitative values given there (in the Abstract) are not good enough to justify widespread medical usage of those assays for diagnosis, as is already acknowledged by the Authors themselves in the paper, at p.2 lines 54-55. Therefore, the statement about the "excellent clinical performance" in the Abstract should be deleted, or -at best- drastically rephrased. This may also require some rephrasing of some of the conclusions as well. Thus, the statement "individuals with fever and respiratory symptoms showed higher viral loads than those without those symptoms" should be explained and justified in more detail, so as to demonstrate a clear dependence, or at least some correlation, which can clearly justify the rapid decrease of performance with the viral load as practically acceptable.
We thank the reviewer for this interesting suggestion and reformulated the sentences abovementioned and those reporting the term “sensitivity”. Focusing on sentence about correlation between symptoms and viral load, we strengthened the statement with addition on another references and details about relationship between higher viral load (in terms of absolute quantification, digital PCR, and Cycle threshold, qPCR) (Lines 178-181).
- Likewise, the term stated as sensitivity in Table 2 is again actually accuracy, and the reference in the text of "diagnostic accuracy" (p.3 lines 103 and 110) should be deleted and the respective sentences rephrased, e.g. "The performance of the LumiraDX .... kit shows an accuracy of ... and a specificity of ...", and similarly the usage of "sensitivity" in other places (e.g. p.3 lines 113 and 115 but also later in other places in the paper which -for brevity- I will not indicate them here one by one) should be sanctioned and correspondingly modified as well.
Modified alongside the text and in the table as suggested.
- The expression of the high viral load which (at p.1 line 24) is first given in percentages (percentages of what?? What is the reference?) should rather be made in terms of the input parameter, i.e. the concentration of SARS-CoV-2 antigens in the blood. This percentage expression is also different from the "viral load" introduced at p.2 lines 83-84, but even here the Authors should briefly define/explain how this viral load is defined and especially measured, in what units?? Also, I personally find it a bit puzzling why the viral load is denoted by "Ct" as none of its letters seem to have any correspondence with the expression "viral load", and by the way shouldn't "t" be a subscript?? Or, in fact, like for many (bio)chemical sensors, "Ct" stands for total concentration (of the measured biocomponent), expressed in, e.g., [milimol/militer]?? Again, the Authors should clarify this aspect as well. Personally, I believe that the Authors should provide an Additional Material (in a separate file, to be added in the Journal together with the main paper) in which they can either briefly describe the definition of all other parameters (e.g. specificity, CI, Cohen's kappa coefficient, k-value, etc.) and/or provide some references and links where (more) such info is provided. This would be especially helpful for readers who are not yet fully familiarized to the domain, with the best example being Ph.D. students who are just starting their work in this field.
Focusing on abstract we specified the reference of percentages: it represents the positive antigenic tests towards the qPCR results, thus the accuracy of the two antigenic tests. Viral load subset (very high, high, moderate) were inferred by qPCR Cycle threshold (Ct, not necessary subscript) since, for all qPCR tests, there is a direct correlation between Ct (comprised between 1 and 45) and increasing of fluorescence. A quantification in terms of copies/ml could be performed if standard were available. However, for SARS-CoV-2 a quantitative evaluation was not necessary for diagnosis since qualitative results were enough.
In our work, we classified samples in abovementioned subsets starting from Ct, which represent the viral load, also according to other studies in literature.
Finally, we agree with reviewer about definition of terms. However, an additional material was, in our opinion, not necessary since these terms are frequently used for statistical analysis. They can be found in many statistical glossaries (an example is https://us.sagepub.com/sites/default/files/upm-assets/41413_book_item_41413.pdf)
- Finally, allow me to make an observation which (at least in my personal opinion, of course) could potentially strengthen significantly the paper. I couldn't help noticing that the paper was very short, and this led me to a more detailed pondering of a few issues which I detail in the next paragraphs. It is unclear for the Reader WHY the Authors chose those specific LumiraDx products. Is it based on cost? or on what? Definitely they cannot be the only products available for the same purpose. I believe that there must be (many) more such products, and then it could be highly relevant for potentially a larger range of Readers, for the Authors to make a similar comparison of a few other representative devices as well. Hence, my suggestion would be for the Authors to add another section, in which they should:
- First, just summarize the (majority of the) most significant products present on the market and available for the same purpose. Of course, some more literature review may also be required to check and present if/what other similar reviews/product comparisons may have been done in literature until now.
- Second/Next, from all those products the Authors should then make a selection of only a few (1, 2, 3? or maybe max.4) of them (i.e. besides those already analyzed in the paper, but whose choice should also be justified, as I mentioned previously). This selection/choice MUST be clearly justified and explained and it must be based -again- on a very clear judgment/selection criterion, e.g.: most popular? most performant? cheapest? Or one (the best) of each category??. Hence a "metric", or a quantitative parameter that describes/characterizes each product of ALL those previously summarized should preferably be defined (or selected from the product's specs) and used to rank it in a list or in more lists (one for each parameter or producer), and then the Authors could presumably select the leader product at the top of each such list. Once these a few other products/devices are thus selected, then the Authors can further carry out the same type of analysis which they did for the LumiraDx kits. Yes, this would mean MUCH more work and more time to spend with experiments and analyses of their results for the benefit of the Readers (eh, noblesse oblige...), but I hope the Authors would agree that such an addition will not only strengthen considerably both the originality and the quality of their paper, but -due to its generality (no longer dealing only with just 2 products from the same manufacturer but with the key products from the major players on the market)- would drastically boost its practical relevance and also its attractiveness for a (much?) larger audience (possibly additionally including more important decisional factors in medical care institutions!).
We agree with this interesting observation. In our hospital the reported tested antigenic assays were used for their logistic and technical features since it is possible, also though a cloud repository, a complete traceability of results, operators, maintenance reports and operational functionality. Moreover, since they are third generation devices, we followed national and international guidelines, which really suggest, if possible, this type of assay as POCT (https://www.ecdc.europa.eu/sites/default/files/documents/Options-use-of-rapid-antigen-tests-for-COVID-19_0.pdf, https://www.amcli.it/wp-content/uploads/2021/03/01-2021_Indicazioni-operative-AMCLI_SARS-CoV-2.v4.pdf). Since no other tests were available in our setting a correlation was not affordable. For this reason, we proposed this manuscript as communication. The main take-home message we want to stress is the need of an in-house validation since no correlation between manufacturer declaration and real clinical performance were generally not observed, also according to several studies. The idea of comparison with other kits is very interesting and we may think to do a future review or metanalysis about this purpose. Nevertheless, an example of in-depth analysis of antigenic test available with their performances can be found to this link: https://www.eurosurveillance.org/content/10.2807/1560-7917.ES.2021.26.44.2100441.
MINOR
- ANY Figure or Table should not be located in the middle of a page and interrupt the fluency of the text, but all Figures/Tables on a page should be placed either at its top and/or its bottom or occupy a whole page.
For position of table and images we followed the journal guidelines, which indicate to allocate them along the manuscript near to section where it was mentioned. However, we tried to allocate them in a suitable position.
- A minor error that may need to be corrected is at p.1 lines 17-18, in the Abstract: replace "non according to manufacturer declarations" with "in disagreement with the manufacturer's specifications".
Modified as suggested.
- A personal suggestion is that some graphics should be added; for many people a drawn graph is more evocative than the numbers in the given Tables. Hence, I suggest the Authors to consider plotting in, e.g., bar charts the data shown in the Tables given in the paper, at least for Tables 1 & 2.
We agree with the reviewer for this very interesting observation. Despite table 1 was not easily plotted, table 2 was reported also in form of bar charts to provide a better comparison about performance of tested assays (figure 1).
All the changes mentioned and reported in the text have been highlighted alongside the manuscript.
Round 2
Reviewer 1 Report
Comments and Suggestions for Authors
The authors have addressed the comments appropriately.